# The cerebellum linearly encodes whisker position during voluntary movement

Susu Chen[1,2,3,4,5], George J Augustine[4,5], Paul Chadderton[1]*

[1]Department of Bioengineering, Imperial College London, United Kingdom; [2]Graduate School for Integrative Sciences and Engineering, National University of Singapore, Singapore; [3]Imperial-NUS Joint PhD program; [4]Lee Kong Chian School of Medicine, Nanyang Technological University, Singapore; [5]Institute of Molecular and Cell Biology, Singapore

**Abstract** Active whisking is an important model sensorimotor behavior, but the function of the cerebellum in the rodent whisker system is unknown. We have made patch clamp recordings from Purkinje cells in vivo to identify whether cerebellar output encodes kinematic features of whisking including the phase and set point. We show that Purkinje cell spiking activity changes strongly during whisking bouts. On average, the changes in simple spike rate coincide with or slightly precede movement, indicating that the synaptic drive responsible for these changes is predominantly of efferent (motor) rather than re-afferent (sensory) origin. Remarkably, on-going changes in simple spike rate provide an accurate linear read-out of whisker set point. Thus, despite receiving several hundred thousand discrete synaptic inputs across a non-linear dendritic tree, Purkinje cells integrate parallel fiber input to generate precise information about whisking kinematics through linear changes in firing rate.

*For correspondence:
p.chadderton@imperial.ac.uk

Competing interests: The authors declare that no competing interests exist.

## Introduction

Tactile sensation is an active process whereby sensory information is acquired through self-initiated movement. Effective sensory processing therefore involves the interplay between motor and sensory systems, incorporating multiple feedback loops (*Diamond et al., 2008*; *Matyas et al., 2010*; *Bosman et al., 2011*). Rodents use coordinated whisker movements for tactile exploration and discrimination, rhythmically sweeping their whiskers back and forth to scan their surroundings. The rodent whisker system thus provides an attractive model to tackle questions related to active sensory processing (*O'Connor et al., 2002*; *Crochet et al., 2011*) and sensorimotor integration (*Kleinfeld et al., 2006*). Amongst multiple processing regions in the brain, the cerebellar cortex is a major site of sensorimotor integration, but little is known about its role in active whisking.

Of the many brain regions involved in whisking behavior, the trigeminal and facial nuclei of the brainstem, thalamus, and neocortex have received the most attention (*Carvell and Simons, 1988*; *Lichtenstein et al., 1990*; *Carvell et al., 1996*; *Fee et al., 1997*; *Kleinfeld et al., 1999*; *Brecht et al., 2004*; *Yu et al., 2006*; *Leiser and Moxon, 2007*; *Herfst and Brecht, 2008*; *Diamond et al., 2008*; *Curtis and Kleinfeld, 2009*; *Hill et al., 2011*; *Crochet et al., 2011*; *Petreanu et al., 2012*). Recently, physiological and anatomical studies have unveiled a whisking central pattern generator located in the reticular formation of the ventral medulla that produces rhythmic signals to muscles that generate whisking (*Moore et al., 2013*). The lateral hemispheres of the cerebellum, in particular lobule Crus I, are strongly implicated in these vibrissae sensorimotor loops (*Shambes et al., 1978*; *Bosman et al., 2011*; *Proville et al., 2014*). Growing evidence suggests synchronization of activity between the cerebellum and other whisker-related brain regions both under anesthesia and during active whisking in the awake state (*O'Connor et al., 2002*;

**eLife digest** Many animals actively move their whiskers back and forth to explore their surroundings and search for objects of interest. This behavior is important for navigation and the animals' sense of touch. It relies on specialized circuits of cells in the brain to carry information about whisker movement patterns and process the touch signals. A region of the brain called the cerebellum is highly connected to these circuits, but its role in the voluntary movement of whiskers is not clear.

Chen et al. aimed to address this question by using a technique called patch clamping to measure the electrical activity of individual neurons in the mouse cerebellum. The experiments revealed that individual cells in the cerebellum called Purkinje cells track whisker movements in real time, and with virtually no delay, through both increases and decreases in their activity. Also, Chen et al. found that the patterns of electrical activity in these cells closely mimicked the positions of the whiskers as they moved. These results tell us that cells in the cerebellum use a simple code to represent whisker position during voluntary movement.

Chen et al.'s findings present the first experimental evidence that the cerebellum applies a type of code known as a linear code to represent the voluntary movements of whiskers. The next challenge is to find out how contact with whiskers alters movement-related signals in the cerebellum.

*Ros et al., 2009*; *Popa et al., 2013*). Sensory-evoked responses are observed in Crus I following whisker stimulation (*Shambes et al., 1978*; *Bower et al., 1981*; *Chadderton et al., 2004*; *Bosman et al., 2010*), and whisker movements can be evoked by optogenetic activation of this lobule (*Proville et al., 2014*), but the principles by which cerebellar neurons encode features of whisking remain to be determined.

In this study, we set out to identify two key aspects of whisking behavior representation in Purkinje cells (PCs), the final stage of information processing and sole output of the cerebellar cortex. Firstly, which kinematic features are represented by PCs? Whisking is a rhythmic process characterized both by fast oscillatory forward and backward movements, as well as slower positional changes (*Hill et al., 2011*). Distinct brain regions make different functional contributions to the encoding of this behavior (*Kleinfeld et al., 2006*). For example, within the neocortex, the phase of whisking is strongly represented in primary somatosensory cortex (vS1, *Curtis et al., 2009*), whereas slower changes (e.g. the set point and amplitude of whisker movement) are more closely correlated with activity in primary motor cortex (vM1, *Carvell et al., 1996*; *Hill et al., 2011*). To understand how the cerebellum fits within the various nested loops of the rodent whisker system, it is necessary to establish the kinematic parameters that are most relevant in the modulation of cerebellar activity. Further, it is important to establish how salient features of whisking are encoded in the activity of single neurons. This information is essential if we are to understand the underlying computational principles of the cerebellar circuit. Changes in both the rate and timing of action potential firing may play a role in sensorimotor encoding (*De Zeeuw et al., 2011*), and it has been proposed that the cerebellum may serve a general function as a linear coding device (*Fujita, 1982*). However, these concepts have not been directly confirmed in the intact brain, and thus the cerebellar coding scheme(s) employed in the representation of whisker movement is unknown.

To address these issues, we have made patch clamp recordings from cerebellar PCs in awake, behaving mice to determine the influence of convergent sensory and motor input on the output patterns of the cerebellar cortex during natural whisking. PCs integrate hundreds of thousands of discrete synaptic inputs (*Palkovits et al., 1977*) across a complex non-linear dendritic tree (*Llinás and Sugimori, 1980a*, *1980b*; *Finch and Augustine, 1998*; *Roth and Häusser, 2001*). Despite this complexity, our experiments demonstrate that single PCs accurately encode ongoing whisker movements via bidirectional modulation of simple spike firing rate in a linear manner. Our results establish the presence of a whisking coordinate system in the cerebellum and reveal the computational algorithm employed during sensorimotor processing.

## Results

### Purkinje cell spiking activity is altered during free whisking

We performed patch clamp recordings in Crus I of cerebellar cortex while using high-speed videography to track the whisker movements of awake head-fixed mice (*Figure 1A–C*). Mice spent variable amounts of time engaged in free whisking, spontaneously switching between periods of whisking and non-whisking behavior (mean whisking bout: 1.68 ± 0.04 s, range: 0.50 – 29.71 s; mean quiet period: 7.82 ± 0.25 s, range: 0.5 – 153.1 s). PCs (n = 70 from 47 mice) could be identified via classification of their two distinct spike waveforms - simple spikes (SS) and complex spikes (CS) - evident in both cell-attached and whole cell recordings (*Figure 1C*).

Although tactile whisker stimulation can change the rate of both SS and CS (*Bower and Woolston, 1983; Loewenstein et al., 2005*; *Bosman et al., 2010*), it is not known whether free whisking also affects PC activity. We therefore compared PC firing rates during epochs of whisking and non-whisking behavior. In the absence of movement, PCs fired SSs at high frequencies (61.9 ± 3.7 Hz, range: 20.1 – 187.9 Hz, n = 70), while the basal rate of CSs was low (1.6 ± 0.1 Hz, range: 0.5 – 3.3 Hz, n = 70). These values are consistent with other recordings made from lateral cerebellar PCs in awake animals (*Fu et al., 1997a*, *1997b*; *Lang et al., 1999*; *Bosman et al., 2010*). During bouts of free whisking, a large proportion of PCs (47 out of 70 cells, ~67%) exhibited significant changes in SS rate (p<0.05, Mann-Whitney-Wilcoxon test). While the majority of PCs (40/47) increased their firing rates (*Figure 1D*, *Video 1*), in some PCs (7/47) SS rates decreased during whisking (*Figure 1E*, *Video 2*). Whisker movement was associated with an overall enhancement of the activity of PCs (*Figure 1F, G*), although the direction and amplitude of SS rate change during whisking was not related to baseline SS firing rates (r = 0.06, p=0.69, n = 47). SS rate changes were therefore non-uniform in both magnitude and sign across the population, with individual PCs exhibiting changes ranging from +128% to -34% during whisking periods (*Figure 1G*). We also compared variability in the timing of simple spiking during periods of whisking and non-whisking behavior by measuring the coefficient of variation (CV) of SS firing. Overall, the CV was close to 1 both when mice were at rest (0.9 ± 0.1, range: 0.3 – 6.2, n = 70), and when they actively moved their whiskers (0.8 ± 0.1, range: 0.3 – 2.5, n = 70). Significant changes in SS variability occurred during free whisking in the majority of PCs that displayed rate modulation (44/47 PCs; see 'Materials and methods'). Moreover, nearly one third (n = 20/70) of PCs demonstrated changes in SS firing regularity alone, suggesting that the temporal patterning of SS might independently encode whisker-related signals. In summary, SS activity – either rate, CV, or both - was altered in nearly all PCs (67/70).

Locomotion can alter whisking behavior: whisker protraction is correlated with running speed (*Sofroniew et al., 2014*). During the course of our PC recordings, mice rarely ran on the treadmill (fraction of all whisking bouts spent running was 3.5%; *Figure 1—figure supplement 1A*). However, in recordings that included periods of locomotion (n = 10), whisker movements were more protracted (*Figure 1—figure supplement 1B, C*) and SS firing rates were elevated, independent of the sign of whisking-related modulation (*Figure 1—figure supplement 1D*; p=0.03, paired t test). These results indicate that running may influence both whisking behavior and whisker-driven SS activity in the cerebellum.

Significant differences in CS firing rates were also observed between whisking and non-whisking conditions in a large proportion of PCs (*Figure 1—figure supplement 2B*; 33 out of 70 cells, p<0.05, Mann-Whitney-Wilcoxon test). A common phenomenon in many regions of the cerebellar cortex is an inverse relationship between SS and CS rate changes during behavior (*Graf et al., 1988*; *De Zeeuw et al., 1995*; *Barmack and Yakhnitsa, 2003*; *Badura et al., 2013*). However, there was no consistent relationship between whisker-related changes in CS and SS frequency (*Figure 1—figure supplement 2B*; r = 0.13, p=0.57). Overall, whisker movements are associated with pronounced alterations in PC activity, indicating that external drive to Crus I changes activity within cerebellar circuits during bouts of whisking.

### Purkinje cell simple spike discharges reliably track whisker movements

To define the relationship between whisker movement and PC simple spiking, we correlated whisker position with the incidence of SSs. Whisking bouts were aligned at their onset to compare the amplitude and duration of whisker movements with firing rate changes both in

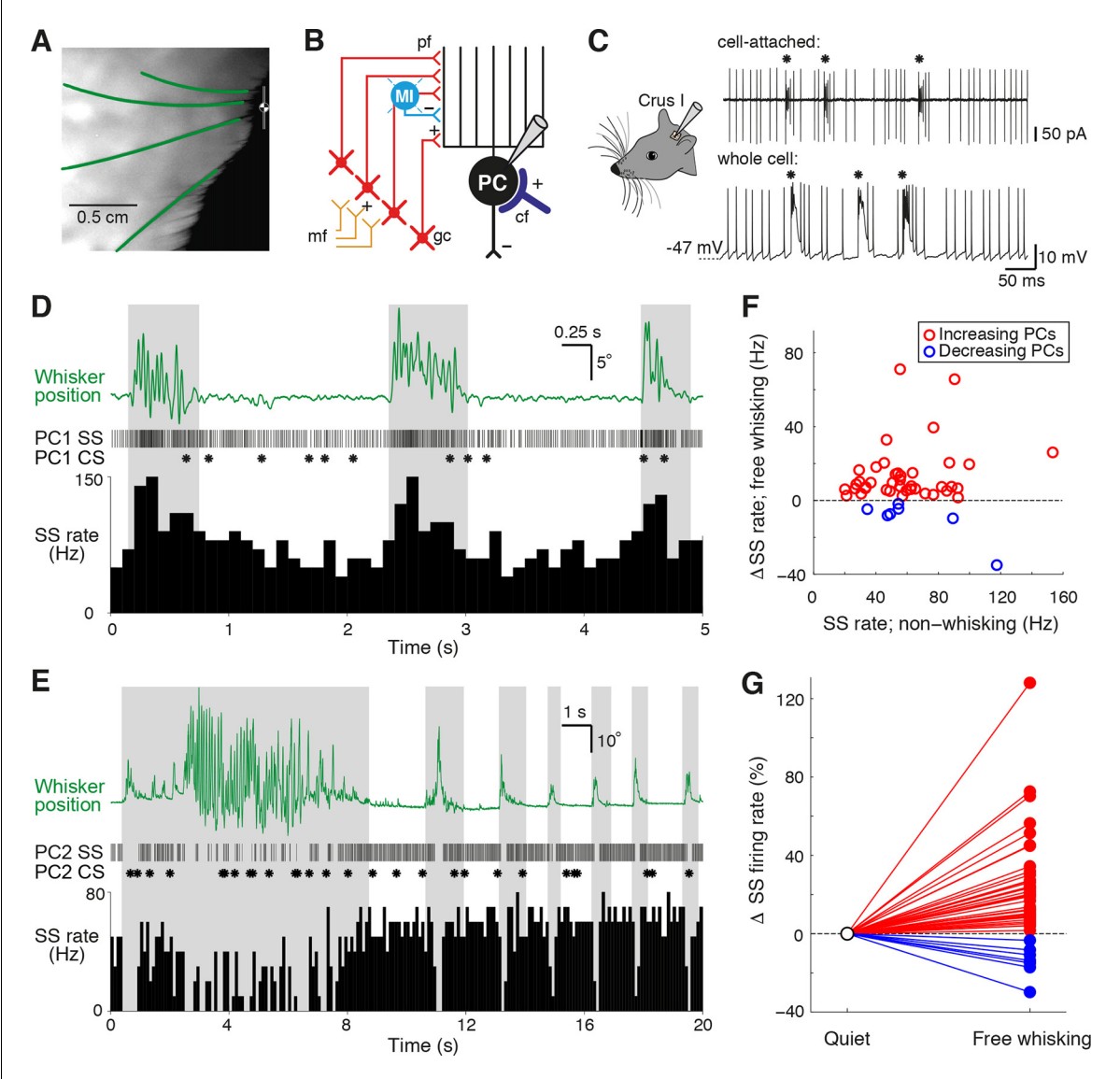

**Figure 1.** Alteration of Purkinje cell activity during free whisking. (**A**) Videography of a head-restrained mouse with four traced whiskers (from row C, labeled in green). (**B**) Simplified diagram of the cerebellar circuit (cf: climbing fiber; gc: granule cell; PC: Purkinje cell; pf: parallel fiber; mf: mossy fiber; MI: molecular layer interneuron). (**C**) PC electrical activity in awake behaving mice, acquired via cell-attached and whole cell patch clamp recordings. Asterisks highlight the incidence of complex spiking. (**D**) Observed behavior of PC that increased simple spike (SS) frequency during spontaneous whisker movements (gray shading), including (top) traced whisker position (green; upward deflections indicate protraction), (middle) corresponding SS and CS trains, and (bottom) SS instantaneous firing rate histogram (bin size: 100 ms). (**E**) Observed behavior of PC that decreased SS frequency during spontaneous whisking. (**F**) Scatter plot showing relative SS firing rate changes during whisking with respect to non-whisking baseline firing rates for all significantly modulated units (p<0.05, n = 47, Mann-Whitney-Wilcoxon test). Red and blue symbols indicate increasing (n = 40) and decreasing (n = 7) PCs, respectively. (**G**) Relative SS firing rate changes with respect to baseline firing rate between quiet wakefulness and free whisking for all modulated cells (red: increasing PCs, blue: decreasing PCs).

The following figure supplements are available for figure 1:

**Figure supplement 1.** Influence of locomotion on simple spike rate alteration during free whisking.

**Figure supplement 2.** Complex spike rate alteration during free whisking.

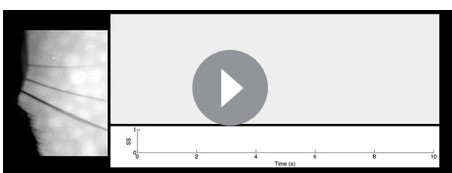

**Video 1.** Increased simple spike activity during whisking. SS activity of a single PC during 15 s of voluntary whisking behavior. Left: Movements of the ipsilateral whisker pad were recorded via high-speed infrared videography. Top: changes in whisker angle for three adjacent row-C whiskers. Bottom and Audio: Simultaneously recorded SS activity from Crus I PC (raster and audio 2x down-sampled for audiovisual clarity).

increasing (*Figure 2A, B*) and decreasing (*Figure 2C*) PCs. Remarkably, the changes in PC SS firing rate during whisking bouts closely mimicked whisker movement (compare upper and lower panels in *Figure 2A*), irrespective of the direction of the rate change (*Figure 2B, C*). Beyond an overall change in CS rate, there was no correspondence between the amplitude/duration of whisker movement and the incidence of CSs (*Figure 2—figure supplement 1*). Therefore, increases and decreases in SS, but not CS, rate are intimately related to changes in whisker position.

## Timing of whisking-related changes in PC firing rates

Because Crus I receives both motor and sensory whisker-related inputs (*Shambes, et al., 1978*; *Bower et al., 1981*; *Proville et al., 2014*; ), SS firing rate changes could reflect either efferent motor command, or re-afferent sensory input. To establish whether changes in PC activity reflect transitions to free whisking in a feed-forward (efferent) or feedback (re-afferent) manner, we determined the temporal relationship between neural activity and behavior. For this purpose, we calculated the cross-correlation between SS firing rate and whisker position, with the analysis centered on the time of whisking onset (*Figure 2D*). Robust correlations between whisker movement and SS discharge were observed with both large and small whisking-related changes in firing rate (*Figure 2—figure supplement 2A,B*), and in a manner that was largely independent of the latency between movement and change in firing rate (*Figure 2—figure supplement 2C*). Two-thirds of PCs (31/47) exhibited SS firing rate changes that preceded whisking onset by a mean of -34.6 ± 7.9 ms (*Figure 2E*), and no clear relationship was observed between the latency and magnitude of whisking related alterations in SS rate (*Figure 2—figure supplement 2D*). Temporal relationships were preserved when the cross-correlation was calculated based on whisking offset (*Figure 2—figure supplement 3*). On average, PC SS discharge led whisker movement by ~18 ms (n = 47) at the population level, implying that SS alterations in Crus I predominantly reflect efferent rather than re-afferent signals. However, the distribution of temporal correlations was broad (range: -178 ms to 324 ms), and non-unimodal (*Figure 2E*, Hartigan's dip test; p<0.001, n = 38), indicating that whisking behavior is represented on multiple timescales amongst neighboring PCs within Crus I. Delayed PC responses relative to whisking may result from additional processing of sensory-driven information and/or recurrent motor-related signals during free whisking. In addition, the transmission of whisker signals via parallel fibers (*Wilms and Häusser, 2015*) of distant granule cells may account for long latency PC responses within the cerebellar cortex.

## Strong linear correlations between SS frequency and whisker position

To establish whether the activity of PCs in Crus I represents salient kinematic parameters of whisking, we directly quantified the relationship between PC SS firing rate and whisker position. Plotting whisker position as a function of SS frequency revealed strong linear correlations: linear regression analysis indicated significant correlations in over 60% of the PCs ($R^2$ = 0.958 ± 0.005, p<0.05, ANOVA, n = 44 ). Two classes of encoding schemes were observed. Unidirectional PCs (*Figure 3A*) displayed linear changes in SS frequency within a range of positions

**Video 2.** Reduced simple spike activity during whisking. SS activity of a single PC during 10 s of voluntary whisking behavior. Left: Movements of the ipsilateral whisker pad were recorded via high-speed infrared videography. Top: Fluctuations in the angle of the C3 whisker. Bottom and Audio: Simultaneously recorded SS activity from Crus I PC (video slowed down 2x for audiovisual clarity).

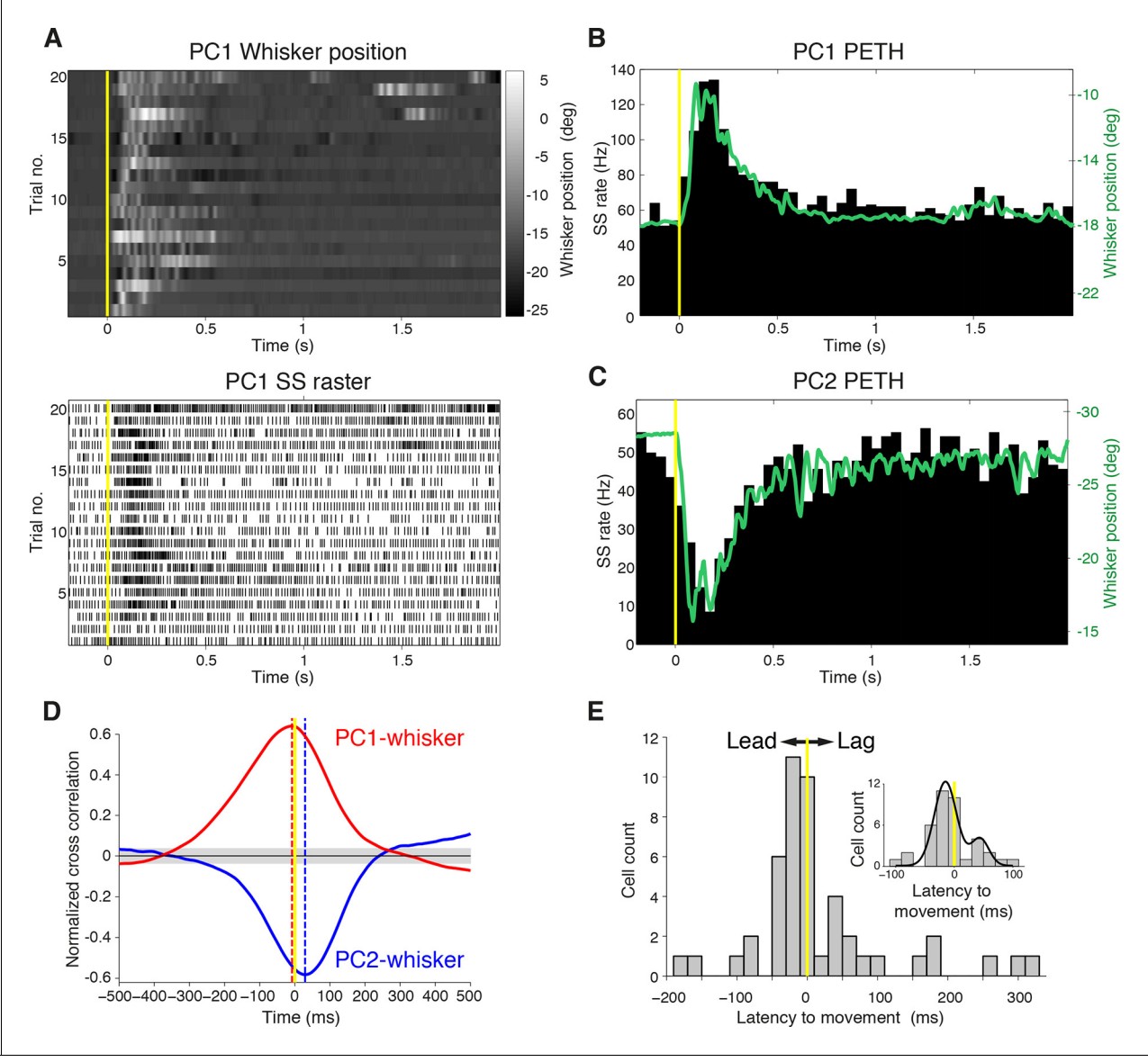

**Figure 2.** Purkinje cell simple spike discharges reliably track whisker movements. (A) Whisker movements and corresponding simple spike raster from a single PC across 20 epochs of free whisking. Neuron demonstrates increased SS frequency during movement. (B) Peri-event time histogram (PETH) for the same PC, obtained by averaging SS rate across trials illustrated in (A), overlaid with averaged whisker position (in green). Note the close relationship between SS firing rate change and mean whisker position. (C) PETH for a PC demonstrating reduced SS frequency during movement. The close relationship between SS firing rate change and mean whisker position is preserved. (D) Normalized cross-correlations between whisker position and SS discharge for exemplar PCs. The peak (red for PC1; shown in A, B) or trough (blue for PC2; shown in C) indicates the temporal relationship between whisker position and spiking. PC1 leads whisker movement by 8 ms (difference between red and yellow dashed lines), while PC2 lags movement by 27 ms (difference between blue and yellow dashed lines). Gray shade demonstrates 95% confidence interval. (E) Temporal relationship between whisker movement and SS discharge for all modulated PCs (bin size: 20 ms). More units show lead (negative latency to movement) than lag (positive latency to movement) with respect to behavior. Inset: zoomed-in histogram between -100 ms and 100 ms. Black line is best fit of two summed Gaussians.

The following figure supplements are available for figure 2:

**Figure supplement 1.** Complex spike relationship to whisker movement.

**Figure supplement 2.** Temporal relationship between whisker movement and SS firing rate for strongly and weakly modulated PCs.

**Figure supplement 3.** Relationship between SS firing rate and whisking offset.

corresponding to only forward or backward movements relative to the whisker resting point (defined as whisker angle during non-movement; see 'Materials and methods'). These were the most common type of response (37/44) and were associated with either increases or decreases in SS rate. In bidirectional PCs, SS frequency both increased and decreased across the full range of movement (*Figure 3B*), with SS frequency encoding both protracted and retracted whisker positions (7/44 PCs). Across the PC population, we observed uni- and bidirectional PCs that encoded whisker position with both positive slopes (corresponding to increased SS rate during forward movements) and negative slopes (decreased SS rate during forward movements). To compare representation of whisker position across the population of PCs, SS firing rate was normalized with respect to the spontaneous firing rate and whisker position was normalized with respect to the whisker resting point. Notably, both classes of PC displayed almost perfect linear relationships (unidirectional: $R^2$ = 0.950 ± 0.004; bidirectional: $R^2$ = 0.959 ± 0.005) between relative SS firing rate change and whisker position (*Figure 3C*). In unidirectional cells, SS frequency at resting point often differed between bouts of movement and non-movement. Bidirectional PCs never exhibited such large baseline shifts in firing frequency and were capable of continuously representing whisker position through bouts of whisking and non-whisking via alteration of SS frequency.

While the gain of SS firing rate changes with respect to whisker position (slope of curves in *Figure 3A–C*) varied across cells (mean = 15.8 ± 2.1 Hz/degree), bidirectional PCs demonstrated significantly higher gain values in comparison to unidirectional PCs (bidirectional: 33.5 ± 4.4 Hz/degree, unidirectional: 12.5 ± 2.0 Hz/degree, p<0.001, Mann-Whitney-Wilcoxon test), indicating that bidirectional PCs are more sensitive to small-amplitude whisker movements (*Figure 3C*, inset). A strong inverse relationship was observed between the linear encoding range and the gain of individual PCs (*Figure 3—figure supplement 1A*) indicating that high-gain PCs linearly encode a relatively smaller portion of all possible whisker angles than low-gain PCs. However, the fraction of time that a whisker spent within a PC's linear encoding regime was approximately constant across the population, with no relationship to PC gain (*Figure 3—figure supplement 1B*). This suggests that both high- and low-gain PCs make similar contributions to the linear encoding of whisker position within the cerebellar cortex. Taken together, these results demonstrate that individual PCs in Crus I represent whisker position linearly and within distinct ranges of movement. The high resting firing rate of individual PCs enables bidirectional representation through both increases and decreases in SS firing rate.

## Cerebellar representation of whisker position does not depend on input from primary motor cortex

In the majority of PCs, whisker-related changes in SS lead, or are coincident with, movement (*Figure 2D*), indicating the dominance of efferent rather than re-afferent drive to the cerebellar cortex. We explored whether the contralateral primary motor cortex, a source of excitatory input to Crus I (*Proville et al., 2014*), could provide an efferent drive during voluntary whisking. We performed patch clamp recordings from PCs in Crus I while locally inactivating contralateral vM1 via muscimol injection (*Figure 3—figure supplement 2A*; see 'Materials and methods'). Seven out of 15 PCs displayed significant SS firing rate modulation (*Figure 3—figure supplement 2B*; p<0.05, Mann-Whitney-Wilcoxon test) with a mean latency to whisker movement of -24.6 ± 11.9 ms (n = 7). Linear representation of whisker position by SS firing rate was preserved irrespective of vM1 inactivation (*Figure 3—figure supplement 2C,D*). Thus, inactivation of vM1 does not degrade the cerebellar representation of whisking, and whisker-related input to the cerebellum is derived from other cortical or subcortical processing stations.

## Most Purkinje cells do not encode phase of whisking cycle via changes in SS activity

Whisking is a rhythmic process and neuronal firing locked to specific phases of the whisking cycle has been observed at multiple processing stations in the brain (*Yu et al., 2006*; *Leiser and Moxon, 2007*; *Curtis and Kleinfeld, 2009*; *Crochet et al., 2011*). To examine whether PCs also encode this parameter, rapidly varying phase information was extracted from whisking bouts by applying the Hilbert transform during epochs of rhythmic whisking (*Figure 4A*). The relationship between simple spiking and phase was then assessed within rhythmic whisking epochs to determine whether PCs

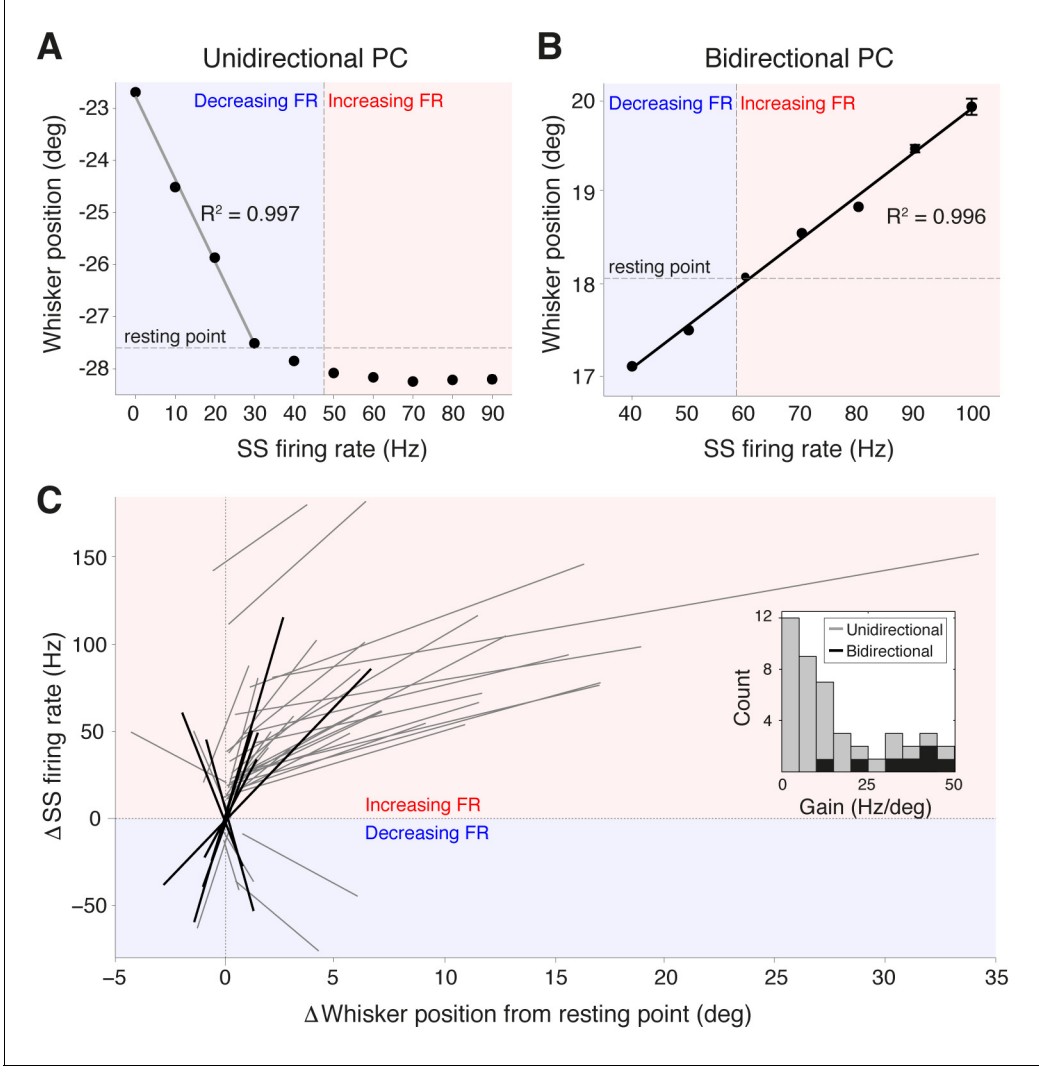

**Figure 3.** Purkinje cell simple spike frequency linearly encodes whisker position. (**A**) Relationship between SS rate and whisker position for PC with strong linear tuning in the forward direction only (unidirectional PC). Linear regression was performed for whisker positions anterior of the resting point (horizontal dashed line). Vertical dashed line shows the cell's spontaneous firing rate (FR). This cell showed linear reductions in SS frequency during forward movement. Blue and red shaded areas represent decreases and increases in SS FR, respectively. (**B**) Relationship between SS rate and whisker position for PC with strong linear tuning in both forward and backward directions (bidirectional PC). Linear fit encompassed the entire range of SS FR modulation. This cell showed increases in SS frequency during forward movement, and decreases in SS frequency during backward movement. (**C**) Summary of all unidirectional PCs (n = 37, gray lines) and bidirectional PCs (n= 7, black lines) with significant linear correlations between SS FR and whisker position (ANOVA, $p<0.05$, $R^2 > 0.86$). FR change and whisker position were normalized with respect to spontaneous firing rate and resting point, respectively. Inset: distribution of gain, defined as the slope of individual linear fit, for both unidirectional (gray) and bidirectional (black) units. Note bidirectional PCs have higher gain values than unidirectional PCs on average, implying they are more sensitive to changes in whisker position during movement.

The following figure supplements are available for figure 3:

**Figure supplement 1.** Linear encoding range of individual PCs.

**Figure supplement 2.** Inactivation of contralateral motor cortex does not degrade cerebellar representation of whisker position.

were more likely to fire at a particular phase. Only PC recordings that coincided with longer periods of exploratory rhythmic whisking were included in this analysis (n = 31, see 'Materials and methods'). In almost every cell (30/31; $p>0.05$, Kuiper test), no significant phase tuning was observed (*Figure 4B,C*). However, one PC did exhibit strong phase preference (modulation depth of 4.8) . In

this cell, a reduction in SS rate was observed during whisking and the incidence of spiking preferentially occurred during whisker retraction (*Figure 4C*). Overall, the vast majority of PCs did not encode rhythmic variations in whisker position within individual whisking cycles, suggesting that phase information is represented by a small fraction of PCs.

## Faithful reconstruction of whisker set point from single Purkinje cell activity

To quantify how well individual PCs represent whisker position, we attempted to reconstruct the trajectory of whisker movement from our recordings of SS activity. A transfer function between SS activity and whisker position was computed from a portion of each recording to capture the underlying linear characteristics of the system (n = 18 PC recordings longer than 300 s, see 'Materials and methods'). Calculated transfer functions were applied to trains of SS activity from the remainder of the recording to test whether the transfer functions could predict the dynamics of whisker movement. Using spike train information from single PCs, it was possible to accurately recover the dynamics of whisker movements over many seconds (*Figure 5A*). In most PCs, the movement reconstruction derived from the transfer function was highly reminiscent of another kinematic parameter – the set point, which denotes the slowly varying midpoint between the protracted and retracted angles of a single whisking cycle. We therefore measured the correlation coefficient between reconstructed whisker position and set point derived from measurements of actual behavior to evaluate the quality of our decoding. Reconstructions based on PC SS activity were excellent predictors of actual whisker set points (*Figure 5B*; range of correlation coefficient: 0.23 – 0.89, mean = 0.57 ± 0.05, p<0.001, n = 18). Therefore, we conclude that single PCs reliably encode whisker set point, and that bidirectional changes in the frequency of simple spiking afford a linear representation of whisker trajectory during voluntary movement.

## Discussion

The rodent whisker system has been extensively studied as a model to understand active sensory processing. However, very little is known about the role of the cerebellum in whisking, despite the established importance of this brain structure for sensorimotor control (*Wolpert et al., 1998*). We have recorded PC activity in behaving mice and, for the first time, have shown that SS firing rates change during active whisker movement. Single PCs estimate slow changes in whisker position in real time via linear alterations in SS frequency to provide accurate information about movement trajectories to downstream neurons.

### Linear coding via non-linear conductances

The computational algorithm by which the cerebellar cortex encodes sensorimotor input has been widely debated. Changes in rate (*Walter and Khodakhah, 2009*) and pauses in firing (*De Schutter and Steuber, 2009*) have been proposed as mechanisms of sensorimotor encoding. While indirect evidence has been found for both schemes, our recordings in behaving animals provide direct support for the proposal that the cerebellar cortex is optimized to perform as a linear coding device (*Fujita, 1982*; *Walter and Khodakhah, 2006*, *2009*). The strong linearity of PC SS output is surprising given the range of non-linear ionic conductances present across the PC dendritic tree (*Llinás and Sugimori, 1980a*, *1980b*) and suggests that non-linear dendritic conductances might compensate for non-linear synaptic integration within passive PC dendrites (*Roth and Häusser, 2001*), or alternatively are disengaged during free whisking. For cerebellar circuits, linear computation may provide the optimal means of performing pattern separation (*Albus, 1971*; *Marr, 1969*; *Walter and Khodakhah, 2009*), affording graded sensorimotor representations (*Heiney et al., 2014*) that are less prone to saturation, and increasing the dynamic range of the system. Both increases and decreases in firing rate are linear with respect to whisker position, suggesting that molecular layer inhibition plays a crucial role in maximizing the range of cerebellar operation (*Park et al., 2012*).

### A whisking coordinate system in lateral cerebellum

The majority of PCs in Crus I represent changes in a simple parameter of whisking behavior, the set point, via SS rate changes. Because of the close and robust correspondence between SS rates and

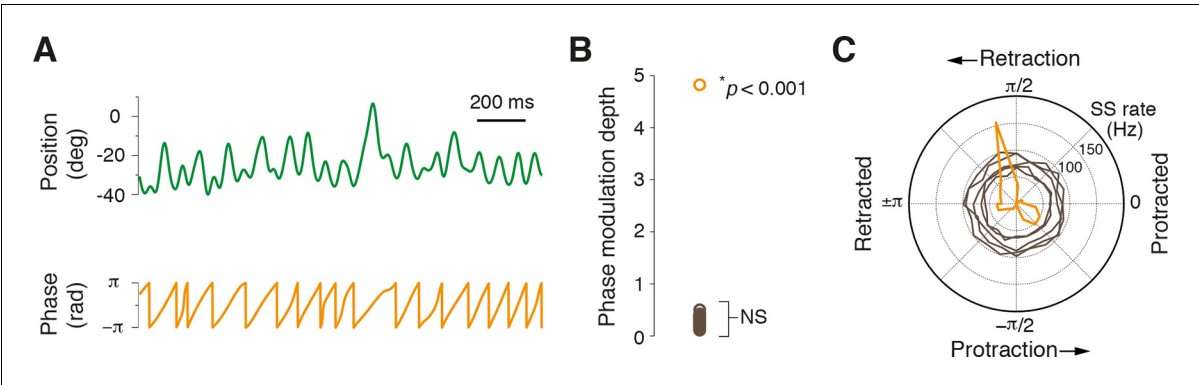

**Figure 4.** Most Purkinje cells do not encode the phase of whisking cycle. (**A**) Example of rhythmic whisker movement (green trace), and corresponding phase (orange) derived from the Hilbert transform of the raw position trace. (**B**) Modulation depth of phase tuning for population of PCs. Phase tuning is absent in SS patterns of all but one PC (NS; not significant, n = 30/31). (**C**) Polar plot depicts the phase tuning of five representative cells that did not demonstrate phase tuning (brown) and one strongly modulated PC (orange), which showed SS firing locked to mid-point of whisker retraction.

movements, we were able to reconstruct whisker trajectories with high accuracy using spike trains from individual cells (*Figure 5*). Several PCs converge onto single target neurons (*Palkovits et al., 1977*) in the deep cerebellar nucleus (DCN), ensuring that set point information is propagated downstream, where integration of PC signals (*Person and Raman, 2012a*) with different rate functions (*Figure 3*) could further extend the dynamic range of DCN neurons to movement. In this arrangement, convergence of positively and negatively modulated PC postsynaptic potentials could degrade set point representation, and therefore it seems likely that these two classes of PC have distinct cellular targets in the DCN (*Person and Raman, 2012b*).

In contrast to set point, phase information is only sparsely encoded in the cerebellar cortex (*Figure 4*). However, phase information may be reconstituted in the DCN via spatiotemporal convergence of weakly tuned PCs (*De Zeeuw et al., 2011*; *Person and Raman, 2012b*). Overall, our data indicate that whisker set point is represented by SS frequency in the majority of Crus I PCs and further suggest that additional phase tuning may restrict SS firing to precise times within the whisking cycle. It is unclear whether the signals we recorded correspond to single or multiple whisker movements. Tactile receptive fields of Crus I PCs can encompass multiple whiskers (*Bosman et al., 2010*), and a similar mapping of motor responses will be helpful to establish how these signals are integrated across the entire cerebellar cortex.

## Dominance of efferent over re-afferent representation of whisking

Neurons in the cerebellar cortex potentially have access to discrete 'motor' and 'sensory' representations of whisking behavior: specifically, efferent copy of planned or current movement from cortical and sub-cortical centers, and re-afferent signals providing continuous sensory feedback from the trigeminal nuclei and sensory neocortex about the consequences of voluntary movement . Necessarily, sensory re-afferent signals are delayed with respect to movement (*Bower et al., 1981*) as they are required to propagate from the periphery, and whisker movements are themselves delayed with respect to muscle activity (by a few tens of milliseconds) owing to inertia (*Berg and Kleinfeld, 2003*). In our recordings, the majority of PCs exhibited changes in SS activity that preceded or were coincident with movement, consistent with a feed-forward as opposed to feedback representation of whisking in the cerebellum. The primary motor cortex does not provide the source of this efferent drive, as the cerebellar representation remains intact during transient vM1 inactivation. Processing stations in the midbrain and the brainstem are therefore likely candidates to provide information to the cerebellum about whisker position during voluntary movement.

## Conclusions

The nervous system retains an internal representation of whisking in the cerebellum via a simple linear encoding regime. This grants a remarkable degree of flexibility to fine-tune and coordinate

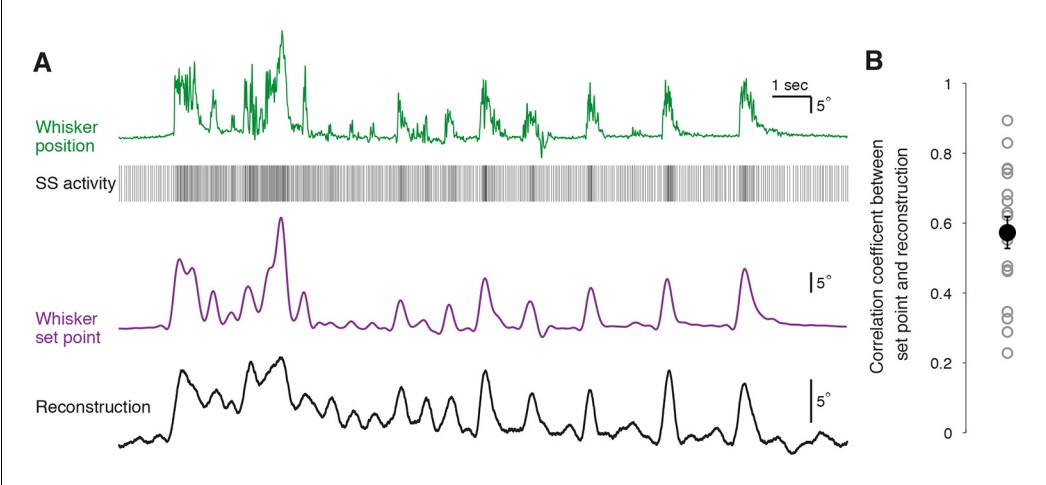

**Figure 5.** Reconstruction of set point trajectories from simple spike activity of single Purkinje cells. (**A**) Reconstruction of whisker movement from single PC SS train based on the calculated transfer function. Whisker set point information (purple) is accurately reconstructed (black trace, bottom) using SS activity from a single PC (down-sampled x3 for visual clarity), highlighting the strong linear relationship between simple spiking and slow whisker kinematics. Correlation coefficient value between reconstruction and set point is 0.78. (**B**) Correlation coefficients between whisker set point and linear reconstruction from individual PCs (gray open circles). Black filled circle: mean ± SEM across all cells (n = 18).

whisker movement by providing fast online feedback, and to disambiguate representations of self- and externally generated sensory signals (*Wolpert et al., 1998*). The resolution with which movement trajectories can be recovered from single neurons suggests that the cerebellar cortex may be an interesting alternative target for brain-machine interface devices that seek to restore movement via online decoding of neural signals (*Kohler et al., 2009*). By providing rapid information about current or future movement, the cerebellar machinery may circumvent long delays in cortical feedback loops, serving effective sensory processing and motor control during active whisking (*Rahmati et al., 2014*). Crucially, our findings confirm that by rendering an internal representation of the whisker system, PC spike train dynamics are highly informative about movement trajectories, facilitating active sensation and tactile exploration.

## Materials and methods

### Surgical procedures and animal handling

The care and experimental manipulation of animals was performed in accordance with institutional and United Kingdom Home Office guidelines. 47 C57BL/6 mice (4–8 weeks old) of both genders were used in this study. Animals were housed in a 12-hr reverse light-dark cycle and all experiments were carried out during the dark phase. Prior to recording, mice were anesthetized with 1–2% isoflurane under aseptic conditions, and a lightweight head-post was attached to the skull using glue (Histoacryl, Braun Corporation, USA) and acrylic dental cement (Kemdent, UK). A circular chamber was built with cement over the lateral hemisphere of the cerebellum to allow subsequent access for electrophysiological recording. A non-steroidal anti-inflammatory drug (Carprofen; 5 mg/kg) was provided via intra-peritoneal administration during surgery to support recovery. Implanted mice were given 2–5 days for recovery, during which time Buprenorphine (0.8 mg/kg) jelly was used for postoperative analgesia. On the day of the recording, mice were first anesthetized with isoflurane (1–2%), and a small craniotomy (1–1.5 mm) was drilled over lobule Crus I. The dura was removed with fine forceps and the craniotomy was covered with 1.5% low-melting point agar and a silicone-based sealant (Kwik-Cast; World Precision Instruments, USA). Ipsilateral whiskers were partially trimmed with one whisker row left untouched (row C or D). At least two hours following these procedures, habituation and recording sessions were started. Mice were carefully placed on a cylindrical treadmill and the head-post was gently loaded into a fixation clamp to painlessly immobilize the head. At least one hour of habituation was allowed for the mice to be acclimated to

the recording environment. Habituated mice showed normal grooming, whisking, and locomotion behaviors on the treadmill. After removal of sealant and agar, recordings were performed in the dark in a single session lasting up to 3 hr.

## In vivo electrophysiology

Whole cell and cell-attached patch clamp recordings were made from cerebellar PCs in awake mice using a Multiclamp 700B amplifier (Molecular Devices, USA). Recordings were made in Crus I (-7 ± 0.2 mm posterior, and 3.4 ± 0.3 mm lateral of bregma) at depths of 350–1500 μm from the pial surface using borosilicate glass pipettes (6–8 MΩ) filled with internal solution containing (in mM): 135 K-gluconate, 7 KCl, 10 HEPES, 10 phosphocreatine, 2 Mg-ATP, 2 $Na_2$-ATP, and 0.5 $Na_2$-GTP (pH 7.2, 280–290 mOsm). Purkinje cells were readily identified by their high spontaneous firing rates and the presence of complex spikes. Data were filtered at 10 kHz and digitized at 25 kHz using an ITC-18 interface (Instrutech Corporation, USA) and acquired on a computer using Axograph X software (www.axograph.com). In whole cell recordings, resting membrane potentials were recorded immediately after formation of whole cell configuration and series resistances ranged between 20 and 40 MΩ. No current was injected and membrane potentials were not corrected for liquid junction potentials.

## Motor cortex inactivation

To transiently inactivate vM1, a small hole was made in the skull above contralateral vM1 (1 mm anterior and 1 mm lateral of bregma) and a guided cannula was inserted 500–600 μm from the pia and fixed with dental cement during head-post implant. The gamma-aminobutyric (GABA) agonist muscimol (ThermoFisher, USA) was administered (0.6 μl of 1μg/μl solution dissolved in 0.9% saline, delivered at a rate of 0.1 μl/min) via a Hamilton syringe into the guide cannula. In pilot experiments, extracellular population recordings (4 shank silicon probe; 4 x 2 tetrode, NeuroNexus, USA) were used to confirm that M1 multi-unit activity was completely abolished within 10 min of injection. For cerebellar recording, injections were made at the end of the habituation session. Electrophysiological recordings began 10 min after infusion and lasted up to 2 hr. Due to the long-lasting effect of muscimol (up to 3 hr), it was not possible to compare single PC activity before and after cortical inactivation.

## Whisker tracking

Under infrared light illumination, whisker movements were filmed with a high-speed camera (Genie HM640; Teledyne Dalsa Inc, USA) operating at 250 frames per second. Video acquisitions were controlled by Streampix 6 software (Norpix, Canada) and externally triggered by TTL pulses generated via the ITC-18 in order to synchronize video and electrophysiological acquisition. Whisker positions were tracked offline using open source software (*Clack et al., 2012*) - http://whiskertracking.janelia.org - and transferred into a graphical user interface in MATLAB (Mathworks) for analysis. Whisker azimuth angles were measured along the medial-lateral axis (medial-lateral line: 0 degree, forward movement: increasing angle, backward movement: decreasing angle); protraction corresponded to increasing whisker angles. Because whiskers, especially those from the same row, move in synchrony, we used one of the traced whiskers for all the analysis, as changing whisker did not affect the results.

We excluded whisker epochs that were shorter than 500 ms and whisker twitches with single back-and-forth deflection smaller than 5 degrees. Whisking epochs were further separated into periods of rest and locomotion. Locomotion episodes were identified as treadmill movement lasting at least 100 consecutive frames (400 ms). The traced whisker position was first low-pass filtered at 30 Hz using a 4-pole Butterworth filter run in forward and reverse directions, and subsequently up-sampled to 1 kHz. Whisking set point was derived by low-pass filtering whisker angle at cutoff frequency 6 Hz. Rhythmic whisking epochs were isolated to determine phase information and cells with >20 s of rhythmic whisker movement were included to evaluate phase tuning. Whisker phase was defined as the angle of the Hilbert transform on band-pass filtered (6–30 Hz) whisker angle. A phase of zero corresponds to maximal protraction and a phase of $\pm\pi$ denotes maximal retraction in a whisk cycle.

## Data analysis

Action potentials were detected offline automatically in Axograph X. SS and CS were sorted according to their distinct waveforms in MATLAB with a manual verification step. The clean separation of SS and CS was confirmed using Peri-CS SS-histograms (*Zhou et al., 2014*). In all PC recordings, histograms showed ~10 ms pauses in SS activity following a CS.

PCs were tested to determine if they exhibited significant changes in SS and CS firing rate between epochs of non-whisking and free whisking in air. On average, 40 episodes of whisking for each cell were used to quantitatively assess how whisking behavior modulated PC firing rate. Spike rates were calculated for individual whisking and non-whisking epochs as the total number of spikes divided by the duration of an epoch. Comparisons of the spike rates were made between quiet epochs and whisking epochs using a Mann-Whitney-Wilcoxon test where $p<0.05$ was recognized as a significant difference. Overall firing rates during whisking and non-whisking were calculated by averaging the spike rates of all epochs comprising the two respective conditions. To generate peri-event time-histograms, spike trains were aligned by the onsets of whisking bouts and averaged across trials. Corresponding whisking epochs were aligned at the onset and averaged to reveal the mean whisker movement within bouts.

Coefficient of variation (CV) of inter-simple spike-interval (ISI) was defined as the standard deviation of ISI divided by its mean, where a $CV > 1$ implies high variance and low regularity. To resolve whether ISI distributions during non-whisking and whisking epochs were significantly different, a 2-sample Kolmogorov-Smirnov test were performed where $p<0.05$ was deemed to be significant.

To determine SS instantaneous firing rates, we used a 100 ms wide rectangular window function and calculated a moving average with 1 ms steps. For cells showing significant SS firing rate modulations during whisking, we truncated spiking and whisker position data into 3-s segments centered on individual whisking onsets/offsets (1 s preceding- and 2 s post-onset/offset). To examine the temporal relationship between SS discharge changes and behavioral transitions, normalized cross-correlations between PC instantaneous firing rate and whisker position were computed for individual data segments and averaged across segments. The time at the nearest maxima/minima (peak/trough) above the upper/lower 95% confidence bounds in the normalized cross-correlation provided the temporal delay between the two signals. To test if the latency distribution was non-unimodal, we computed Hartigan's Dip test on the empirical distribution from -100 ms to 100 ms (n = 38 cells) with 5 or 10 ms bin size.

To determine the directionality of PCs in encoding whisker position, a firing directionality ratio was calculated based on the normalized linear fitting curve for each cell by dividing the minimal value of firing rate change on one of the two directions (increase or decrease) by the other, giving an index value between zero and one. PCs with a ratio value of zero (n = 37 cells) were classified as unidirectional cells, whereas PCs with positive values were classified as bidirectional (n = 7; ratio range: 0.4–0.9).

To identify whether PCs SS rates were significantly modulated by the circular whisking variable phase, a 2-sample Kuiper test ($p<0.05$) was used to compare the distribution of phase information at all times with its distribution at spike times. We divided the phase information into 20 bins and calculated a histogram of the spike events. This histogram was then normalized by the amount of time spent in individual bin to generate values concerning firing rate. The modulation depth of phase was computed as the maximal firing rate minus the minimal firing rate divided by the mean firing rate.

All data are presented as mean ± SEM unless otherwise stated.

## Acknowledgements

We thank Claudia Clopath and Troy Margrie for feedback on the manuscript, Ian Duguid and Marta Jelitai for advice and support with awake patch clamp recordings, and Daniel Hill for generous help with implementing the transfer function. This work was supported by a NUS Graduate School scholarship to SC, a CRP grant from the National Research Foundation of Singapore to GA, and a MRC Career Development Award to PC (G1000512).

# Additional information

## Funding

| Funder | Grant reference number | Author |
| --- | --- | --- |
| Medical Research Council | Career Development Award, G1000512 | Paul Chadderton |
| National Research Foundation Singapore | CRP grant | George J Augustine |
| National University of Singapore | Graduate School Scholarship | Susu Chen |

The funders had no role in study design, data collection and interpretation, or the decision to submit the work for publication.

## Author contributions

SC, Conception and design, Acquisition of data, Analysis and interpretation of data, Drafting or revising the article; GJA, Conception and design, Drafting or revising the article; PC, Conception and design, Analysis and interpretation of data, Drafting or revising the article

## Author ORCIDs

Paul Chadderton, http://orcid.org/0000-0002-6276-1011

## Ethics

Animal experimentation: The care and experimental manipulation of animals was performed in accordance with the United Kingdom Animals (Scientific Procedures) Act (1986). Protocols were approved by the Animal Welfare and Ethical Review Board of Imperial College London and the UK Home Office, and were conducted under Home Office Project Licence (PPL 70/7205). All surgery was performed under general anesthesia, and every effort was made to minimize suffering.

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
