## [Decision Letter]

Thank you for submitting your work entitled "The cerebellum linearly encodes whisker position during voluntary movement" for peer review at *eLife*. Your submission has been favorably evaluated by Gary Westbrook (Senior editor), a Reviewing editor, and three reviewers, one of whom, Karel Svoboda (Reviewer 2), has agreed to reveal his identity.

Both the reviewers and the Reviewing editor find considerable merit in this work. The claim that cerebellar spiking may determine the set point is a novel and strong claim and perhaps, given some qualms about other issues, this part should receive greater emphasis. As the reviewers note, there are many issues that need to be settled, some that may require additional experiments.

1) The impact of locomotion on the cerebellar simple spiking needs to be clarified. As noted by the reviewers, in principle, this could be solved by comparing whisking during resting and during locomotion. In particular, it may well be that whisking touch signals are linked to locomotion through contact of the vibrissae in synchrony with each step.

2) The issue of timing of neuronal activity related to the onset of whisking is not properly dealt with, so that conclusions that this represents efferent signals (from M1?) versus afferent signals (from S1? brainstem?) are unfounded. As it stands, these claims need to be downplayed. Simple experiments, like transient inactivation M1 could be added.

3) The issue of control of the set-point needs to respect the bounded extent of whisking. It would be important to plot the change in whisker position as a function of the starting position. At starting positions near full protraction or retraction one would expect a contraction of the range. This issue may relate back to (1), as there is some evidence for protractions with running speed (Sofroniew and Svoboda 2015).

Note that the reviewers have discussed the reviews with one another and the Reviewing editor has drafted this decision to help you prepare a revised submission. Please address the issues raised by the reviewers upon submission of a revised manuscript; the full set of comments from the reviewers are included below.

Reviewer #1:

This is an important contribution to the field because few active whisking studies have focused on cerebellum and because the findings are intriguing. The results do not definitely address the issue of efference versus re-afference but this can be done in subsequent work. I have a couple of points that the authors may be able to clarify. About 67% of cells showed SS activity related with whisking. Did the authors check the receptive fields of the cells and does this explain the lack of responsiveness in the remaining group?

Reviewer #2:

Although whisker movements have been extensively studied in the context of active sensation, the control of whisker position is not understood. The vibrissal motor cortex (vM1) has access to the facial muscles (e.g. low threshold for evoking movements with u-stimulation), but coding of variables related to whisker movement is weak and sparsely distributed. Also, I am not aware of behavioral experiments showing that vM1 is required for purposive whisker movements. Kleinfeld's group has made headway in identifying the cpg for rhythmic whisker movements, but this is likely only part of the story. The cerebellum is a core structure in sensorimotor integration and likely plays a role in control of whisker movements, but few measurements have investigated coding of whisker movement related variables in the cerebellum. This is where the present study makes a significant contribution. The authors find a strong representation of whisker set point in the simple spike (SS) rate of Purkinje neurons in Crus 1. Virtually every PC codes for the whisking set point (the position over which whiskers move back and forth). This is an interesting result. I have some concerns with key aspects of the secondary analysis and the framing of the study.

1) The authors argue that the cerebellar coding reflects an efferent signal. This conclusion is almost entirely based on latency analysis. First, latency arguments are dangerous. How is whisking onset defined on a trial-by-trial basis? This is critical because the threshold determined the relative timing of whicker movement and changes in SS activity! Whisker movements might lag the earliest re-afferent signals by 10 ms or more (because of inertia and drag whisker movements are expected to be delayed with respect to compression of the facial musculature). Second, the arguments based on known physiology in vS1 and vM1 and cortico-cerebellar interactions are likely off-base. The cerebellum receives short-latency sensory input and interacts with other motor-related structures in the brain stem. I would be surprised if interactions with vM1 or vS1 play important roles in the phenomena studied here. In addition, these hypothesis are easy to test. Simply transiently inactivate vS1 or vM1 and see what happens to the code! Third, millisecond time resolution requires analysis at the level of average PSTHs. A latency analysis needs to be done on a trial-by-trial basis. Why not do this using membrane potential?

2) Not clear if the classification into bidirectional and unidirectional makes a lot of sense. The authors need to show that the data really fall into two classes or form a continuum.

3) I don't understand the business about non-linear dendrites and linear encoding. The mystery is unclear. There is no modeling that deals with this issue.

4) It would be nice to learn how widespread the representation of whisking setpoint is.

Reviewer #3:

The authors address the correlation between Purkinje cell firing in Crus 1 and active whisker movements. They found that a large number of Purkinje cells shows (complex and/or simple spike) firing rate modulation during voluntary whisker movements. While the role, if any, of the complex spike rate modulation remains rather elusive, the authors found that simple spike firing is closely related to the set point of the whisker, with a linear encoding of the whisker angle. Although a few recent articles addressed this topic, the role of the cerebellum in the rodent whisker system remains largely unclear. This well-designed and executed study confirms the implication of cerebellar Purkinje cell activity in the whisker system. A number of points require attention before this manuscript could be accepted for publication:

1) Locomotion:

In this study, the mice are placed on a treadmill. There is quite some evidence that locomotion and whisking are tightly correlated. Nevertheless, the authors do not address this issue. To what extent do the authors think that locomotion-related changes in firing could be a (potentially) confounding factor?

2) Simple spike rate modulations (Figure 1):

Although the approach taken by the authors to identify modulated Purkinje cells seems to be ok, they report Purkinje cells with minimal changes in simple spike firing (<5 Hz) as being significantly modulated. It would be good to give an indication of the variations during quiet and active phases, as well as in between rest and activity.

3) Latency to movement (Figure 2):

The authors compared the instantaneous SS firing rate by a sliding 100 ms window with the whisker movement. This approach yields a wide range of latencies, with a tendency of PCs to start SS modulation just before the onset of movement. I have three questions on this approach:

How constant were these latencies between different whisker bouts when recording from a single Purkinje cell?

And: some cells showed minimal SS firing rate modulation (see remark 2). Do the authors think that with such minimal modulation a reliable latency could be found? Is it possible to indicate the relation between modulation strength and latency?

And finally: was there also a relation between the end of a whisking bout and SS firing?

4) Whisker "resting point" and linear schemes (Figure 3):

Here the authors show that there is a linear relation between SS firing and whisker position. Whisker can move over a long range (see Figure 1, for example). This is a bit puzzling to me: if cells can modulate up to 50 Hz/degree, than their coding range has to be limited with respect to the tens of degree that a whisker can move. Second, the whisker resting position can change between different periods of rest (at least: in our hands it does). Did the authors observe a similar phenomenon and was this also reflected in their data?

---

## [Author Response]

Both the reviewers and the Reviewing editor find considerable merit in this work. The claim that cerebellar spiking may determine the set point is a novel and strong claim and perhaps, given some qualms about other issues, this part should receive greater emphasis. As reviewers note, there are many issues that need to be settled, some that may require additional experiments.

1) The impact of locomotion on the cerebellar simple spiking needs to be clarified. As noted by the reviewers, in principle, this could be solved by comparing whisking during resting and during locomotion. In particular, it may well be that whisking touch signals are linked to locomotion through contact of the vibrissae in synchrony with each step.

We have separated our electrophysiological data into epochs of running and non-running behavior in order to resolve the impact of locomotion on Purkinje cell simple spiking. In general, our mice did not run very much during the course of recordings. Across our entire data set, mice ran for only 3.5% of whisking bouts (11/47 recordings; a total 94.6 s), so the vast majority of whisker-related modulation in SS rate was measured in non-running mice. We are therefore confident that the correlation between SS rate and whisker position is not a consequence of synchrony between locomotor and vibrissal movements. We have included two video sequences (Video 1 and Video 2) that show the alteration of Purkinje cell SS activity during whisking bouts, in the absence of locomotion.

In a subset of recordings, we did observe locomotion during bouts of whisking, and here we have quantified the impact of running on whisking-related SS firing rates. This analysis suggests that firing rates in Crus I are elevated during locomotion. We have included this information in a new figure (Figure 1—figure supplement 1) and discuss these points in the Results (subsection “Purkinje cell spiking activity is altered during free whisking”, third paragraph).

2) The issue of timing of neuronal activity related to the onset of whisking is not properly dealt with, so that conclusions that this represents efferent signals (from M1?) versus afferent signals (from S1? brainstem?) are unfounded. As it stands, these claims need to be downplayed. Simple experiments, like transient inactivation M1 could be added.

We have undertaken additional analysis as suggested (see specific responses to Reviewers 2 and 3). We have also performed experiments to pharmacologically inactivate contralateral vibrissal motor cortex (via muscimol injection), in order to test the proposal that vM1 provides the source of efferent drive to the cerebellar cortex. These results indicate that motor cortex does not provide this drive, as inactivation of vM1 does not obviously degrade the cerebellar representation of whisker position. We have added these results to a new figure supplement (Figure 3—figure supplement 2), and have substantially revised our Discussion regarding the timing and origin of whisker-related signals in the cerebellum on the basis of these new data.

3) The issue of control of the set-point needs to respect the bounded extent of whisking. It would be important to plot the change in whisker position as a function of the starting position. At starting positions near full protraction or retraction one would expect a contraction of the range. This issue may relate back to (1), as there is some evidence for protractions with running speed (Sofroniew and Svoboda 2015).

We have addressed this issue by quantifying the range of whisker positions over which Purkinje cells linearly encode whisker position. We find a strong inverse relationship between the linear encoding range and gain of individual PCs. Therefore, high-gain neurons encode a relatively smaller proportion of overall whisking space than low-gain neurons. Interestingly, we have also quantified the fraction of time that whiskers spent within the linear encoding regime, and we found no correlation with PC gain. This suggests that PC gain may be set adaptively to match the ongoing dynamics of whisker movement. We have included this information in a new figure (Figure 3—figure supplement 1) and in the related part of the Results (subsection “Strong linear correlations between SS frequency and whisker position”, second paragraph).

Reviewer #1:

This is an important contribution to the field because few active whisking studies have focused on cerebellum and because the findings are intriguing. The results do not definitely address the issue of efference versus re-afference but this can be done in subsequent work. I have a couple of points that the authors may be able to clarify. About 67% of cells showed SS activity related with whisking. Did the authors check the receptive fields of the cells and does this explain the lack of responsiveness in the remaining group?

Unfortunately we were not able to measure the receptive fields of individual cells during the course of our recordings. As all recordings were performed in unanaethetised animals, we were unable to get close to the whisker pad and systematically test responses to individual whiskers with a stimulator (e.g. a piezo wafer) without potentially disturbing the animal. In the absence of active touch, whiskers move in synchrony (see Video 1; also Hill et al. 2011), therefore we are able to relate SS rate changes to whisking without necessarily knowing the identity of the whisker(s) providing input. The functional organization of whisker input in the cerebellum has been previously described: in Crus I and II, 75% of cells showed detectable changes in SS firing rate with tactile stimulation (Bosman et al. 2010). With this in mind, we suggest that between a quarter and third of Crus I PCs do not receive prominent whisker input, which is consistent with the fraction of cells that do not show whisking-related changes in SS rate. We now specifically address this point in the Discussion (subsection “A whisking coordinate system in lateral cerebellum”, last paragraph).

Reviewer #2:

Although whisker movements have been extensively studied in the context of active sensation, the control of whisker position is not understood. The vibrissal motor cortex (vM1) has access to the facial muscles (e.g. low threshold for evoking movements with u-stimulation), but coding of variables related to whisker movement is weak and sparsely distributed. Also, I am not aware of behavioral experiments showing that vM1 is required for purposive whisker movements. Kleinfeld's group has made headway in identifying the cpg for rhythmic whisker movements, but this is likely only part of the story. The cerebellum is a core structure in sensorimotor integration and likely plays a role in control of whisker movements, but few measurements have investigated coding of whisker movement related variables in the cerebellum. This is where the present study makes a significant contribution. The authors find a strong representation of whisker set point in the simple spike (ss) rate of Purkinje neurons in Crus 1. Virtually every PC codes for the whisking set point (the position over which whiskers move back and forth). This is an interesting result. I have some concerns with key aspects of the secondary analysis and the framing of the study.1) The authors argue that the cerebellar coding reflects an efferent signal. This conclusion is almost entirely based on latency analysis. First, latency arguments are dangerous. How is whisking onset defined on a trial-by-trial basis? This is critical because the threshold determined the relative timing of whicker movement and changes in ss activity! Whisker movements might lag the earliest re-afferent signals by 10 ms or more (because of inertia and drag whisker movements are expected to be delayed with respect to compression of the facial musculature).

We agree that measurement of latency can be problematic, especially as we are monitoring spontaneous movements rather than externally triggered behavior. For the peri-event time histogram plots, whisking onset was identified by frame-by-frame analysis of images of whisker position. However, to determine the temporal relationship between changes in whisker position and SS rate, we performed cross-correlation analysis to measure the average delay between these two signals. We have analyzed the variability of our latency measurement, using individual epochs of whisking and find that we can reliably measure the temporal delay between the two signals in both strongly and weakly modulated PCs (Figure 2—figure supplement 2; see our response to Reviewer 3).

We accept the point about inertia and drag on whisker movements, which we had not considered previously, and now refer to this directly in the Discussion (subsection “Dominance of efferent over re-afferent representation of whisking”).

Second, the arguments based on known physiology in vS1 and vM1 and cortico-cerebellar interactions are likely off-base. The cerebellum receives short-latency sensory input and interacts with other motor-related structures in the brain stem. I would be surprised if interactions with vM1 or vS1 play important roles in the phenomena studied here. In addition, these hypothesis are easy to test. Simply transiently inactivate vS1 or vM1 and see what happens to the code!

We have done these additional experiments, and confirm that we were indeed off-base. We recorded Purkinje cell activity following pharmacological inactivation of vM1 (via muscimol injection) to test whether motor cortex plays a causal role in the representation of whisker position in the cerebellum. We recorded the activity of 15 Purkinje cells in 9 mice following vM1 inactivation, and observed significant whisker-related changes in firing rate in 7 cells. In 6 of these cells, we measured a clear linear relationship between SS rate and whisker position. We include this information in a new supplementary figure (Figure 3—figure supplement 2), and have now reformulated our discussion about the potential sources of whisker drive to the cerebellum (Results subsection "Cerebellar representation of whisker position does not depend on input from primary motor cortex”; and Discussion subsection “Dominance of efferent over re-afferent representation of whisking”). We are grateful to the reviewer for raising the issue and suggesting this course of action!

Third, millisecond time resolution requires analysis at the level of average PSTHs. A latency analysis needs to be done on a trial-by-trial basis. Why not do this using membrane potential?

In our recordings, Purkinje cells were spontaneously active at high rates, with membrane potential essentially resting at threshold. As a result, we were unable to use membrane potential changes to determine movement onset, even with averaging.

2) Not clear if the classification into bidirectional and unidirectional makes a lot of sense. The authors need to show that the data really fall into two classes or form a continuum.

To determine the directionality of PCs in encoding whisker position, a firing directionality ratio was calculated based on the normalized linear fitting curve for each cell by dividing the minimal value of firing rate change on one of the two directions (increase or decrease) over the other, giving an index value between zero and one. We find that cells fall into two classes: PCs with a ratio value of zero (n = 37 cells) were classified as unidirectional cells, whereas PCs with positive values were classified as bidirectional (n = 7; range for bidirectional cells: 0.4 - 0.9). This information is now contained in the Materials and methods (subsection “Data analysis”, fourth paragraph).

3) I don't understand the business about non-linear dendrites and linear encoding. The mystery is unclear. There is no modeling that deals with this issue.

We apologize for the lack of clarity. The crux of our point is that Purkinje cells encode a key kinematic parameter in a strikingly linear manner. This means that whisker-driven synaptic input is integrated in such a way as to produce a linear spike rate modulation. Much in vitro work has revealed non-linear conductances in the dendrites and soma of Purkinje cells (e.g. Ca^2+^ activated potassium conductances, I_h_, recurrent Na^+^ conductance), so we are highlighting that these conductances interact (or less likely, are not active) during whisking in order to maintain linear encoding. Our aim was not to explicitly model this phenomenon, but rather provide evidence from the behaving animal for a proposal that has been previously presented (e.g. Walter and Khodakhah, 2006, 2009). We have amended the text in order to clarify this point in the manuscript (Introduction, third paragraph).

4) It would be nice to learn how widespread the representation of whisking setpoint is.

We were not able to systematically map the location of set point-encoding PCs in this study, due to the difficulties of utilizing patch clamp recording for micro-mapping. However, all of our recordings were located in Crus I, with the coordinates -7 ± 0.2 mm posterior, and 3.4 ± 0.3 mm lateral of bregma. We have added this information to the Materials and methods (subsection “In vivo electrophysiology”).

Reviewer #3:

The authors address the correlation between Purkinje cell firing in Crus 1 and active whisker movements. They found that a large number of Purkinje cells shows (complex and/or simple spike) firing rate modulation during voluntary whisker movements. While the role, if any, of the complex spike rate modulation remains rather elusive, the authors found that simple spike firing is closely related to the set point of the whisker, with a linear encoding of the whisker angle. Although a few recent articles addressed this topic, the role of the cerebellum in the rodent whisker system remains largely unclear. This well-designed and executed study confirms the implication of cerebellar Purkinje cell activity in the whisker system. A number of points require attention before this manuscript could be accepted for publication:1) Locomotion:In this study, the mice are placed on a treadmill. There is quite some evidence that locomotion and whisking are tightly correlated. Nevertheless, the authors do not address this issue. To what extent do the authors think that locomotion-related changes in firing could be a (potentially) confounding factor?

To address this point, we have split our whisking bouts into periods of running and non-running. During most of our recordings, mice did not run, preferring to remain stationary on the treadmill for long periods. In total, locomotor bouts constitute only 3.5% of time spent whisking. Given the very small fraction of time mice spent running in our study, we do not believe that locomotion-related changes in firing are a significant confounding factor in our measurement of the relationship between whisking and SS activity. However, in 10 recordings, where we had at least 5 s of running combined with whisking, we were able to measure the influence of locomotion on SS rate changes.

Both the magnitude of SS rate alterations and the distribution of whisker angles were altered during locomotion. This indicates that locomotor signals can exert a direct influence on whisker-related SS activity in Crus I. Firing rates in Crus I were elevated during locomotion, and this was the case regardless of whether PCs showed increasing or decreasing responses during whisking alone. We have added a new supplementary figure (Figure 1—figure supplement 1) to include this information, and address the potential contribution of locomotor signals in the Results (subsection “Purkinje cell spiking is altered during free whisking”).

2) Simple spike rate modulations (Figure 1):

Although the approach taken by the authors to identify modulated Purkinje cells seems to be ok, they report Purkinje cells with minimal changes in simple spike firing (<5 Hz) as being significantly modulated. It would be good to give an indication of the variations during quiet and active phases, as well as in between rest and activity.

We have now quantified variability within and between non-whisking and whisking epochs and include this information in the Results as follows:

“We also compared variability in the timing of simple spiking during periods of whisking and non-whisking behavior by measuring the coefficient of variation (CV) of SS firing. Overall, the CV was close to 1 both when mice were at rest (0.9 ± 0.1, range: 0.3 – 6.2, n = 70), and when they actively moved their whiskers (0.8 ± 0.1, range: 0.3 – 2.5, n = 70). Significant changes in SS variability occurred during free whisking in the majority of PCs that displayed rate modulation (44/47 PCs; see Materials and methods). Moreover, nearly one third (n = 20/70) of PCs demonstrated changes in SS firing regularity alone, suggesting that the temporal patterning of SS might independently encode whisker-related signals.”

3) Latency to movement (Figure 2):

The authors compared the instantaneous SS firing rate by a sliding 100 ms window with the whisker movement. This approach yields a wide range of latencies, with a tendency of PCs to start SS modulation just before the onset of movement. I have three questions on this approach:

How constant were these latencies between different whisker bouts when recording from a single Purkinje cell?

We measured the latency between movement and spiking using cross-correlation for individual whisking bouts, and now include this information as part of a new figure (Figure 2—figure supplement 2). While we observe some variability in the latencies between different bouts, our metric accurately captures the average temporal relationship between movement and spiking. Given that other sensorimotor input is likely to influence Purkinje cell activity in Crus I, it is unclear where the source(s) of this variability may arise.

And: some cells showed minimal SS firing rate modulation (see remark 2). Do the authors think that with such minimal modulation a reliable latency could be found?

We compared the distribution of latency estimates (calculated from individual whisking bouts) for strongly and weakly modulated Purkinje cells, and have included two example plots, and summary data in the new figure supplement. We find examples of similar variability for strongly and minimally modulated neurons and therefore conclude that our latency measures can accurately capture the temporal relationships of weakly modulated PCs. Overall, we have tried to address the issue in the manuscript by displaying the full range of measured values, i.e., we plot the relationship of latency with respect to peak coefficient value and modulation strength for the entire population (Figure 2—figure supplement 2; see also next point). Weakly modulated and/or longer latency responses during whisking may be expected in a subset of PCs given that the anatomical arrangement of parallel fibers could lead to propagation and dispersion of signals in time and space.

Is it possible to indicate the relation between modulation strength and latency?

We include this information in the same figure supplement – overall, we do not observe a strong correlation between modulation strength and latency.

And finally: was there also a relation between the end of a whisking bout and SS firing?

We generated cross-correlograms using whisking bouts centered on offsets and compared them with those generated previously using whisking bouts centered on onsets (Materials and methods). No significant differences were observed, and thus we conclude that the relationship between SS spiking and whisking is maintained throughout bouts of movement. This information is now included in Figure 2—figure supplement 3 and in the Results (subsection “Timing of whisking-related changes in PC firing rates”).

4) Whisker "resting point" and linear schemes (Figure 3):

Here the authors show that there is a linear relation between SS firing and whisker position. Whisker can move over a long range (see Figure 1, for example). This is a bit puzzling to me: if cells can modulate up to 50 Hz/degree, than their coding range has to be limited with respect to the tens of degree that a whisker can move.

Individual PCs exhibited considerable variability both in their gain, and the absolute range of whisker angles over which they were linear encoders. We looked into this issue by plotting gain vs. encoding range, and found a strong inverse relationship (R^2^ = 0.75) between the two. Therefore, high-gain PCs encode only a small range of whisker angles. This could indicate that high-gain PCs make a relatively smaller contribution to the representation of whisker position than low-gain PCs. However, when we quantified the fraction of time that a whisker spent within a PC’s linear encoding regime, we found that this was roughly constant across the population, with no relationship to PC gain (R^2^ = 0.05). In other words, we observed high-gain PCs when behavior was characterized by bouts of low amplitude whisker movement, and low-gain PCs when behavior was characterized by large movements. These data hint that the gain of whisker encoding may be subject to adaptive changes either within, or upstream of the cerebellar cortex. We have included this information in a new figure (Figure 3—figure supplement 1) and have discussed it in the Results (subsection “Strong linear correlations between SS frequency and whisker position“, second paragraph).

Second, the whisker resting position can change between different periods of rest (at least: in our hands it does). Did the authors observe a similar phenomenon and was this also reflected in their data?

We observed small changes in resting position. The average change was 7.1 ± 0.8 degrees across all of our recordings (n = 44). We looked to see if the change in resting position was reflected in the SS rate of neurons with high gain (>30 Hz/degree). In 4 out of 9 cells we found a significant linear correlation with resting position (p<0.05, ANOVA).